# Quantification of southwest China rainfall during the 8.2 ka BP event with response to North Atlantic cooling

**Y. Liu**[1,2] **, C. Hu**[1]

[1] State key lab of biogeology and environmental geology, China University of Geosciences, Wuhan, 430074, PR China

[2] Faculty of Materials Science & Chemistry, China University of Geosciences, Wuhan, 430074, PR China

*Correspondence to*: Y. Liu (yhliu@cug.edu.cn)

Abstract. The 8.2 ka BP event could provide important information for predicting abrupt climate change in the future. Although published records show that the East Asian monsoon area responded to the 8.2 ka BP event, there is no high resolution quantitative reconstructed climate record in this area. In this study, a reconstructed 10-yr moving average annual rainfall record in southwest China during the 8.2 ka BP event is presented by comparing two high-resolution stalagmite $\delta^{18}O$ records from Dongge cave and Heshang cave. This decade-scale rainfall reconstruction is based on a central-scale model and is confirmed by inter-annual monitoring records, which show a significant positive correlation between the regional mean annual rainfall and the drip water annual average $\delta^{18}O$ difference from two caves along the same monsoon moisture transport pathway from May 2011 to April 2014. Similar trends between the reconstructed rainfall and the stalagmite Mg/Ca record, another proxy of rainfall, during the 8.2 ka BP period further increase the confidence of the quantization of the rainfall record. The reconstructed record shows that the mean annual rainfall in southwest China during the central 8.2 ka BP event is less than that of present (1950 ~ 1990) by ~200 mm, and decreased by ~350 mm in ~70 years experiencing an extreme drying period lasting for ~50 years. Comparison of the reconstructed rainfall record in southwest China with Greenland ice core $\delta^{18}O$ and $\delta^{15}N$ records, suggests that the reduced rainfall in southwest China during the 8.2 ka BP period was coupled with Greenland cooling with a possible response rate of $110 \pm 30$ mm/$℃$.

## 1 Introduction

As evidence in support of global warming becomes stronger, it is apparent that the anticipated rise in sea levels may be higher than expected (Rahmstorf, 2007) and the frequency and amplitude of abrupt climate change (Martrat et al., 2004; Pall et al., 2007) may also be greater. As climate events are likely to be problematic for both ecosystems (Walther et al., 2002) and human society (Khasnis and Nettleman, 2005), any aid in prediction is crucial.

Studies of past climate events could hopefully provide useful information for exploring trigger mechanisms (Cheng, et al., 2009; Liu et al, 2013). The 8.2 ka BP event is noted to be the most pronounced abrupt climate event occurring during the whole Holocene (Alley and Ágústsdóttir, 2005). The highest magnitude variation across the low to high latitudes makes a viable target for numerical modelings (Daley et al, 2011; Morrill et al., 2011) and may offer an insight into the sensitivity of

climate response in different areas (Condron and Winsor, 2011; LeGrand and Schmidt,
2008). This event was firstly identified in Greenland ice cores (Alley et al., 1997),
showing a duration of 160-yr (Thomas et al, 2007) with a temperature drop of 3.3±1.1℃
in central Greenland (Kobashi et al, 2007), and is known globally (Dixit et al., 2014;
Morrill et al., 2013; Ljung et al., 2008; Ellwood and Gose, 2006). However, as most
records associated with this event mainly derived from North Atlantic and Europe
(Daley et al., 2011; Szeroczyńska and Zawisza, 2011; Snowball et al., 2010; Hede et
al., 2010; Dom ńguez-Villar et al., 2009; Prasad et al., 2009), the question remains as
to how much it influenced the East Asian monsoon area (EAMA).
Although some proxies from lake sediments (Yu et al., 2006; Hong et al., 2009;
Zheng et al., 2009;Mischke and Zhang, 2010), stalagmites (Wu et al, 2012; Cheng et
al., 2009; Hu et al., 2008a; Wang et al., 2005; Dykoski et al., 2005) and marine
sediments (Zheng et al., 2010; Ge et al., 2010) do record the 8.2 ka BP event in
EAMA, only Hu et al. (2008a) attempted a quantitative reconstruction of rainfall by
using stalagmite $\Delta\delta^{18}O$ records which indicated a decrease in precipitation during the
event in southwest China, an area influenced by East Asian monsoon. However, the
resolution of this precipitation record is approximately 100-yr and needs to be
improved.
Based on the method presented by Hu et al. (2008a) , this study reconstructs a 10-yr
averaged annual rainfall record in southwest China during the 8.2 ka BP event by
comparing sub-annual (Liu et al., 2013) and 3.5-yr resolution stalagmite $\delta^{18}O$ (Cheng
et al., 2009) records from the same moisture transport pathway. This study further
addresses the sensitivity of the climate of southwest China to North Atlantic cooling
during the 8.2 ka BP event, providing quantitative data for simulating this global
event in climate system models.
## 2 Methods
### 2.1 Rainfall reconstruction
It has been previously discussed (Hu, et al., 2008a) that, in a monsoon area, regional
rainfall histories could be reconstructed by using coeval stalagmite $\delta^{18}O$ comparisons
between two close sites located along the same atmospheric moisture transport
pathway, as the difference allows the removal of secondary controls, such as moisture
transport and temperature on $\delta^{18}O$. Working with this premise, two published high
resolution stalagmite $\delta^{18}O$ sequences during the 8.2 ka BP event from Heshang cave,
central China (Liu et al., 2013) and Dongge cave, southwest China (Cheng et al.,
2009), located directly upstream in the atmospheric pathway(Fig.1), were investigated.
### 2.1.1 Stalagmite $\Delta\delta^{18}O$ sequence establishment
There is only one high-resolution $\delta^{18}O$ record from stalagmite HS4 in Heshang cave
(30 °27′N, 110 °25′E)(Fig. 1), central China, covering the 8.2 ka BP period (Liu et al.,
2013), with an average resolution of ~0.3-yr. However, there are two published
stalagmite $\delta^{18}$O records (stalagmite DA and D4) from Dongge cave (25°17′N,
108°5′E)(Fig.1), southwest China (Wang et al., 2005; Dykoski et al., 2005). Cheng et
al.(2009) re-dated both DA and D4 from Dongge cave across the 8.2 ka BP period to
produce a better controlled chronology, giving $\delta^{18}$O records with an average
resolution of ~3.5-yr and ~2-yr (Cheng et al., 2009) respectively. These records are
then compared with HS4 using the approach outlined in Hu et al. (2008a).
Fig. 2 shows the $\delta^{18}$O records from HS4 (Fig. 2a) (Liu et al., 2013), DA (Fig. 2b)
and D4 (Fig. 2c) (Cheng et al., 2009) where similar structural patterns are observed
with matching major peaks and troughs. Corresponding peaks or troughs are marked
as shown by dashed lines in Fig. 2 and the chronology of DA, D4 and HS4 are so
matched to reduce the chronological uncertainty. It should be noted that the wiggle
matching is within the analytical uncertainty of the U-Th chronology. As the
measurement resolutions of HS4, DA and D4 are different, all sequences were first
processed to create records of equivalent annual resolution allowing the resultant time
sequences to be used to construct a 10-yr moving average sequence. Two $\delta^{18}$O
difference ($\Delta\delta^{18}$O) sequences between HS4 and adjusted DA records (Fig. 2d) and
between HS4 and adjusted D4 records (Fig. 2e) were thus established.
Though there is a systematic offset between Fig. 2d and Fig. 2e, the variations and
trends of the two sequences are similar, suggesting either of the two $\Delta\delta^{18}$O sequences
could be used for the following reconstruction. Since the $\delta^{18}$O record from Dongge
cave adopted in Hu et al.(2008a) is from DA, this study also uses data from DA (Fig.
2d) for further rainfall reconstruction.
## 2.1.2  Uncertainties of $\Delta\delta^{18}$O
The use of $\Delta\delta^{18}$O to reconstruct regional rainfall requires some understanding of the
uncertainties within the measurement records and calculations. The U/Th dating
maximum uncertainty of stalagmite DA during the 8.2 ka BP period is ~90-yr (Cheng
et al., 2009), while the average difference between the adjusted and original DA data
set to match HS4 is ~40-yr. This adjustment is within the dating uncertainty, but to
test the robustness of the approach, the whole DA $\delta^{18}$O data set is shifted by 50-yr in
both older and younger directions and the resultant data sets are compared. These
$\Delta\delta^{18}$O sequences are shown in Fig. 3a along with unchanged DA chronology (black),
shifting DA 50-yr younger (blue) and 50-yr older (red). Fig. 3a suggests that though
the time shifted data sets show increased variability of the $\Delta\delta^{18}$O with a maximum
error of 0.76‰, the general variation trends are similar, indicating that this difference
method is sufficiently robust for this study.
In addition to the chronology uncertainty, other factors may affect the accuracy of
$\Delta\delta^{18}$O. $\delta^{18}$O analytical uncertainties of the HS4 and DA datasets are 0.08‰ (Liu et al.
2013) and 0.15‰ (Cheng, et al., 2009) respectively. Additionally the standard
deviation of the 10-yr average, especially the largest standard deviation of $\Delta\delta^{18}$O
between DA and HS4 is 0.62‰. And an estimated uncertainty of 0.35‰ from the
model established by Hu et al. (2008a) should be noted. Taking all of these factors
into consideration, the final uncertainty of the $\Delta\delta^{18}O$ sequence during the 8.2 ka BP
period is estimated to be ~0.53‰.

### 2.1.3 Rainfall reconstruction

Based on the $\Delta\delta^{18}O$ sequence shown in Fig. 3b, the rainfall during the 8.2 ka BP
period is reconstructed using the previous model presented by Hu et al. (2008a) via
the relation between $\Delta\delta^{18}O$ and rainfall (Rainfall=189.08×$\Delta\delta^{18}O$ +1217.4) (Hu et al.,
2008a). The uncertainties from $\Delta\delta^{18}O$ give an error of ~100 mm/yr for the
reconstructed rainfall record.
Reconstruction of regional rainfall using two spatially separated cave records on
the same moisture transport pathway requires stalagmite $\delta^{18}O$ values from monsoon
areas to faithfully preserve rainfall information. Stalagmite $\delta^{18}O$ values are influenced
by different types of precipitation, along with the source and pathways of moisture,
plus local condensation and evaporation processes (Dayem et al., 2010). A recent
millennial scale climate simulation(Liu et al., 2014) suggests that Chinese stalagmite
$\delta^{18}O$ records might be useful as indicators of intensity of the East Asian summer
monsoon in terms of the continental scale Asian monsoon rainfall response in the
upstream regions. As both Dongge and Heshang $\delta^{18}O$ records respond to upstream
rainfall, the difference of the two records is expected to directly reflect the rainfall
between Dongge and Heshang.
On decadal scale, relationships between $\Delta\delta^{18}O$ and rainfall have previously been
discussed (Hu et al., 2008a), and we here attempt to utilize this approach on an inter-
annual time scale. A direct test of the validity of using moisture transport pathways
would use cave drip water $\delta^{18}O$ ($\delta^{18}O_d$) signals from both DA and HS4 sites which
should reflect speleothem $\delta^{18}O$ variations directly. Unfortunately there is no published
monitored data from Dongge. However there is some recently published $\delta^{18}O_d$ data
from a cave named Liangfeng (26°16'N, 108°03'E)(Fig. 1) from May 2011 to April
2014 with local precipitation $\delta^{18}O$($\delta^{18}O_p$) data (Duan et al., 2016). Liangfeng is close
to Dongge(25°17'N, 108°5'E)(Fig.1) and may therefore be an alternative data source
to assess the validity of the rainfall reconstruction method.
There are three separate sequences of $\delta^{18}O_d$ from different drip sites in Liangfeng
cave (Zeng et al, 2015). From them, LF6 with the lowest drip rate but highest
variation has been selected, as it is considered to record climate information more
efficiently. The lowest drip rate of LF6 from Liangfeng cave suggests fresh water is
being mixed with delayed transit stored water for this drip site (Zeng et al, 2015) and
may provide longer term instead of short seasonal information compared with the
other two sites. Based on the 3 years of monthly monitored data of LF6 and HS4
(Duan et al., 2016), the sequences of annual moving average $\delta^{18}O_d$ of LF6 and HS4
have been calculated. Generally LF6 $\delta^{18}O_d$ values are higher than those of HS4, which
is sensible since Heshang cave is further along the moisture transport pathway(Fig.1).
LF6 $\delta^{18}O_d$ is considered mixed fresh and stored water, so its response to the local
rainfall is expected to be delayed. A calculated positive correlation($R^2$=0.62) between
the local annual moving average $\delta^{18}O_p$ and the 2-month delayed annual moving
average of LF6 $\delta^{18}O_d$ (monthly $\delta^{18}O_p$ and $\delta^{18}O_d$ data are from Duan et al., 2016)
strongly suggests that a delay of 2 months should be applied when using LF6 $\delta^{18}O_d$
data. A similar analysis of $\delta^{18}O_d$ at HS4 site and $\delta^{18}O_p$ outside Heshang cave (Duan et
al., 2016) reveals a positive correlation($R^2$=0.71) between the local annual moving
average $\delta^{18}O_p$ and a 4-month delayed annual moving average for HS4 $\delta^{18}O_d$.
Combined analysis shows a positive correlation ($R^2$=0.72) between the 2-month
delayed annual moving average of LF6 $\delta^{18}O_d$ and the 4-month delayed annual moving
average of HS4 $\delta^{18}O_d$, giving some support to the idea that the controlling factors on
both LF6 and HS4 $\delta^{18}O_d$ are similar.
After the time adjusted $\Delta\delta^{18}O_d$ sequence has been built, the correlation between
$\Delta\delta^{18}O_d$ and the regional average annual rainfall at six sites between Dongge cave and
Heshang cave detailed in Hu et al. (2008a)(Fig. 1) may be compared. The regional
average annual rainfall is calculated from monthly instrumental records between May
2011 and April 2014 from http://www.wunderground.com/history/. Fig. 4 shows that
there is a significant positive correlation ($R^2$=0.79) between annual average $\Delta\delta^{18}O_d$
and regional annual rainfall, supporting the idea that the stalagmite $\Delta\delta^{18}O$ between
two caves located along the same moisture transport pathway can provide information
on regional rainfall variation.
## 2.2 Mg/Ca data processing
In addition to $\Delta\delta^{18}O$, the Mg/Ca ratio, another important rainfall proxy, can be
considered. The Mg/Ca data set is taken from Liu et al. (2013) measured using a
JEOL JXA8800R Electron Microprobe at the Department of Material Sciences,
Oxford, along the HS4 stalagmite growth axis. The Mg/Ca data were processed to
provide annual resolution and a 10-yr moving average constructed in the same way as
for $\delta^{18}O$.
## 3  Results
The 10-yr moving average $\Delta\delta^{18}O$ record from DA and HS4 is shown in Fig. 3b.  It is
reasonable that the DA $\delta^{18}O$ values are generally higher than those of HS4 (Fig. 2a
and Fig. 2b) since Heshang Cave is located further along the moisture transport
pathway(Fig. 1) and is so expected to displayed a systematic $\delta^{18}O$ offset. The average
$\delta^{18}O$ difference between HS4 and DA is 1.0‰ during the whole Holocene (Hu, et al,
2008), while the average $\Delta\delta^{18}O$ value during the 8.2 ka BP event shown in Fig. 3b is
much lower at 0.26‰.
During the central event, it is notable that some of the $\Delta\delta^{18}O$ values are around
zero or even negative, indicating much reduced moisture transport during that time.
While the lowest value of $\Delta\delta^{18}O$ is nearly -0.50‰ (Fig. 3b), we do not expected
negative $\Delta\delta^{18}O$ values. The estimated uncertainty of ~0.53‰ in the $\Delta\delta^{18}O$ detailed
in section 2.1.2, along with difference in evaporation in the two caves is likely to
contribute to producing a negative $\Delta\delta^{18}O$. Cave monitored data suggest evaporation
may occur during dripping and enhanced processes in a dry season could result in
heavier drip water $\delta^{18}O$ values (Zeng et al., 2015), especially in a well ventilated cave.
Dongge is a cave consisting of branches with twists and turns, while Heshang is a
much simpler cave with a nearly straight main passage, and a 20 m high entrance (Hu
et al., 2008b). Heshang cave is clearly more open and better ventilated than Dongge
cave leading to greater heat and moisture exchange between the inside and outside
cave (Hu et al, 2008b). During similar dry conditions, the evaporation effects in
Heshang cave are expected to be more significant than in Dongge Cave, and the drier
the condition, the heavier HS4 $\delta^{18}O$ values expected, leading to lower or even
negative $\Delta\delta^{18}O$ values between DA and HS4. Thus less rainfall is still related to lower
$\Delta\delta^{18}O$ values. Since the 8.2 ka BP event is the driest period during the whole
Holocene (Hu et al., 2008), negative $\Delta\delta^{18}O$ values produced during the central event
are possible.
From the 10-year moving average $\Delta\delta^{18}O$ obtained from HS4 and DA records(Fig.
3b), there is a significant change in value from 1.3‰ to -0.5‰ over approximately 70
years at the commencement of the event. Compared with the average amplitude of
$\Delta\delta^{18}O$ during the whole Holocene of 1.0‰ (Hu et al., 2008a), this is a surprisingly
large change.
From the $\Delta\delta^{18}O$ record shown in Fig. 3b, using the previously determined relation
(Rainfall=189.08$\times\Delta\delta^{18}O$ +1217.4) from Hu et al. (2008a), the rainfall record in
southwest China during the 8.2 ka BP period may be established as shown in Fig.3b.
While some support for the reconstruction method can be obtained using recent
monitoring records detailed in section 2.1.3, stalagmite Mg/Ca ratios also provide
some useful corroborative information.
Stalagmite Mg/Ca ratio is a proxy mainly controlled by local rainfall with higher
Mg/Ca values corresponding to lower rainfall (Fairchild and Treble, 2009), though it
may show some temperature dependence, increasing slightly with temperature. The
variation is understood to result from $CO_2$-degassing occurring earlier during water
movement in dry seasons as cave water seeps more slowly, thus Ca is lost from karst
waters by formation of calcite earlier during transport processes and before waters
reach the stalagmite. Such a prior-calcite-precipitation process would be expected to
produce higher Mg/Ca ratios (Tremaine and Froelich, 2013; Fairchild and Treble,
2009). Although it is hard to obtain quantitative rainfall data from Mg/Ca ratios, the
variation of Mg/Ca may give a qualitative indication of rainfall variability and trend.
Therefore the variation of Mg/Ca ratios could indicate whether the reconstructed
rainfall from $\Delta\delta^{18}O$ is reliable or not.
The HS4 Mg/Ca sequence presented as a 10-yr moving average record during the
8.2 ka BP event is shown in Fig. 3c. As high Mg/Ca values are considered to indicate
low rainfall, the Y axis of Mg/Ca was reversed to make the comparison clearer. Both
the Mg/Ca ratios and the reconstructed rainfall data are presented as 10-yr moving
average values. Although the two data sets show slight differences, there is a general
inverse relationship between the two sequences giving a correlation coefficient ($R^2$) of
0.56 (n=219). And overall similarity could be observed between the trends of the two
patterns with high (low) Mg/Ca values corresponding to low (high) rainfall, which
suggests that the Mg/Ca results generally support the reconstructed rainfall record.

The reconstructed rainfall record (Fig. 3b) shows a maximum decline in annual
rainfall of 350 mm/yr, which is nearly twice that obtained from the low-resolution
(~100-yr) rainfall record (Hu et al., 2008a) during the same period and the lowest
annual rainfall in this study is lower than that from Hu et al. (2008a) by ~100 mm.
This is believed to be a result of the record resolution. Fig. 3b also shows that the
period of decreasing rainfall at the beginning of the event lasts for ~70 years, before
entering into an extreme dry period. During the central period of the 8.2 ka BP event,
the average annual rainfall is only ~1200 mm, which appears to be the driest period
during the whole Holocene in this area, lasting for ~50 years. The rainfall calculation
developed in Hu et al. (2008a) was made by averaging annual rainfall records from 6
sites between Heshang and Dongge(Fig. 1) and the averaged annual rainfall between
1950 and 1990 from the 6 sites is ~1380 mm, indicating the average annual rainfall
during the central 8.2 ka BP period is less than present by ~200 mm.
## 4  Discussions
It has been reported that the response of the EAMA to North Atlantic cooling during
the 8.2 ka BP event results from atmospheric rather than oceanic processes (Liu et al.,
2013). It might be assumed that the high northern latitude ice-cover reinforces
Northern Hemisphere cooling, increasing the temperature gradient between the high
and low latitudes which leads to southward migration of the inter-tropical
convergence zone (Chiang and Bitz, 2005; Broccoli et al., 2006). This would result in
weakening of the East Asian Monsoon and increased aridity. Assessment of the
sensitivity of southwest China climate response to North Atlantic cooling might
provide a clue to how North Atlantic cooling affects the EAMA.

Fig. 5 demonstrates three sequences of Greenland ice core $\delta^{18}$O (Thomas et al.,
2007)(Fig. 5a) , a palaeo-temperature indicator (Stuiver, et al., 1995), Greenland ice
core $\delta^{15}$N (Kobashi et al., 2007)(Fig. 5b), a newly developed palaeo-temperature
proxy (Buizert et al., 2014) and the reconstructed rainfall record in southwest China
during the 8.2 ka BP period(Fig. 5c). The data shown in Fig. 5a are from Thomas et al.
(2007) with a 3-yr resolution. To allow comparison with the reconstructed rainfall
records, the $\delta^{18}$O of the ice core was processed to provide a 10-yr moving average.
The $\delta^{15}$N data in Fig. 5b are from Kobashi et al. (2007) with a 11-yr resolution and
were processed similarly.

As low Greenland ice $\delta^{18}$O and $\delta^{15}$N values indicate local cooling (Thomas et al.,
2007; Kobashi et al., 2007), both Fig. 5a and Fig. 5b reveal similar trends of
decreasing temperature during the 8.2 ka BP event. The comparison between each
data set in Fig. 5 suggests that the decrease in rainfall in southwest China may indeed
be in response to Greenland cooling. Further analysis shows a weak-correlation
between Greenland ice core $\delta^{18}$O and the reconstructed rainfall with a correlation
coefficient ($R^2$) of 0.47 (n=219) indicating a 1‰ drop in Greenland ice core $\delta^{18}$O
could lead to ~7% decrease in rainfall in southwest China. Though there is not enough
$\delta^{15}N$ data to reveal further correlations, it does indicate a drop of 3.3±1$^{\circ}C$ when the
8.2 ka BP event occurred(Kobashi et al., 2007). As the reconstructed annual rainfall
record reveals a maximum decrease of 350 mm, the magnitude of rainfall response of
southwest China to Greenland cooling during 8.2 ka BP period could be assessed as
110±30 mm/$^{\circ}C$.

## 5  Conclusions

1. Based on a comparison of two high-resolution stalagmite $\delta^{18}O$ records from
Dongge cave and Heshang cave along the monsoon moisture transport
pathway in China, a 10-yr moving average quantitative annual rainfall record
in southwest China is established during the 8.2 ka BP event.
2. Significant positive correlation between recent monitored drip water annual
average $\delta^{18}O$ differences from two caves along the monsoon moisture
transport pathway and the regional average annual rainfall from May 2011 to
April 2014 provides support for the reconstruction. Similar trends between the
reconstructed rainfall and stalagmite Mg/Ca ratios, another proxy of rainfall,
increase the confidence of the quantization of the rainfall record.
3. The reconstructed rainfall record shows that the annual rainfall in southwest
China decreased sharply by ~350 mm in ~70 years when the 8.2 ka BP event
occurred and experienced an extreme drying period lasting for ~50 years
during the central event. Compared with the modern instrumental records, the
averaged annual rainfall in southwest China during the 8.2 ka BP event is less
than that of present (1950 ~ 1990) by ~200 mm.
4. A comparison between reconstructed rainfall in southwest China and
Greenland ice core $\delta^{18}O$, an indicator of temperature, suggests that the rainfall
decrease in southwest China during the 8.2 ka BP period coupled with
Greenland cooling. A possible response rate of 110±30 mm/$^{\circ}C$ could be
presumed by the temperature drop derived from Greenland ice core $\delta^{15}N$ and
rainfall decrease from the reconstructed record.
**Acknowledgements.** This work was supported by NSFC Grants 41371216 and
41130207. We thank the editor, Prof. Dominik Fleitmann, and the two anonymous
reviewers for their valuable comments that greatly improved the manuscript. We also
thank Dr. Nick Belshaw for proofreading the manuscript.

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

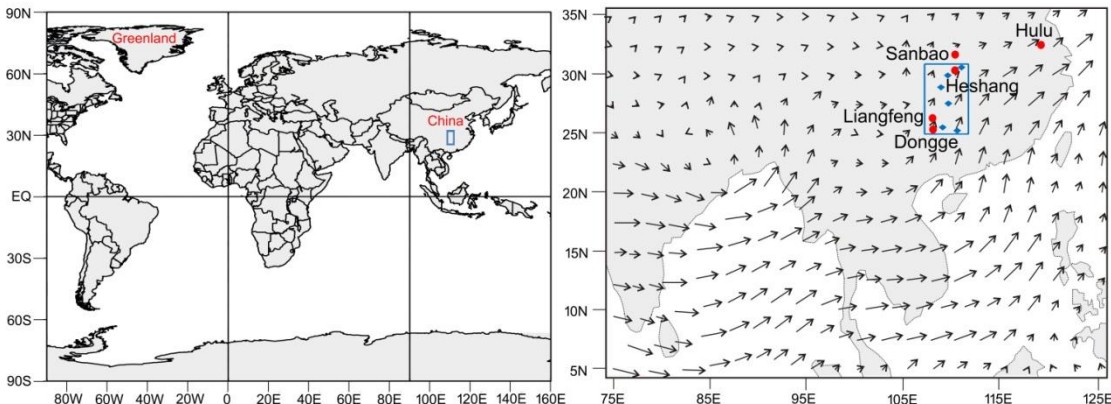


Figure 1. Location maps. The left map shows the location of Greenland and southwest China(blue box). The right map shows the location of Heshang cave and other Chinese caves(red points), with main feature of the summer monsoon marked. Smaller arrows reflect moisture transport and direction averaged over the whole atmosphere (Ding et al., 2004). The blue box indicates the specific region for which comparison of Heshang and Dongge allows rainfall reconstruction, and the blue diamond patterns show the location of six modern rainfall stations detailed in Hu et al.(2008) .

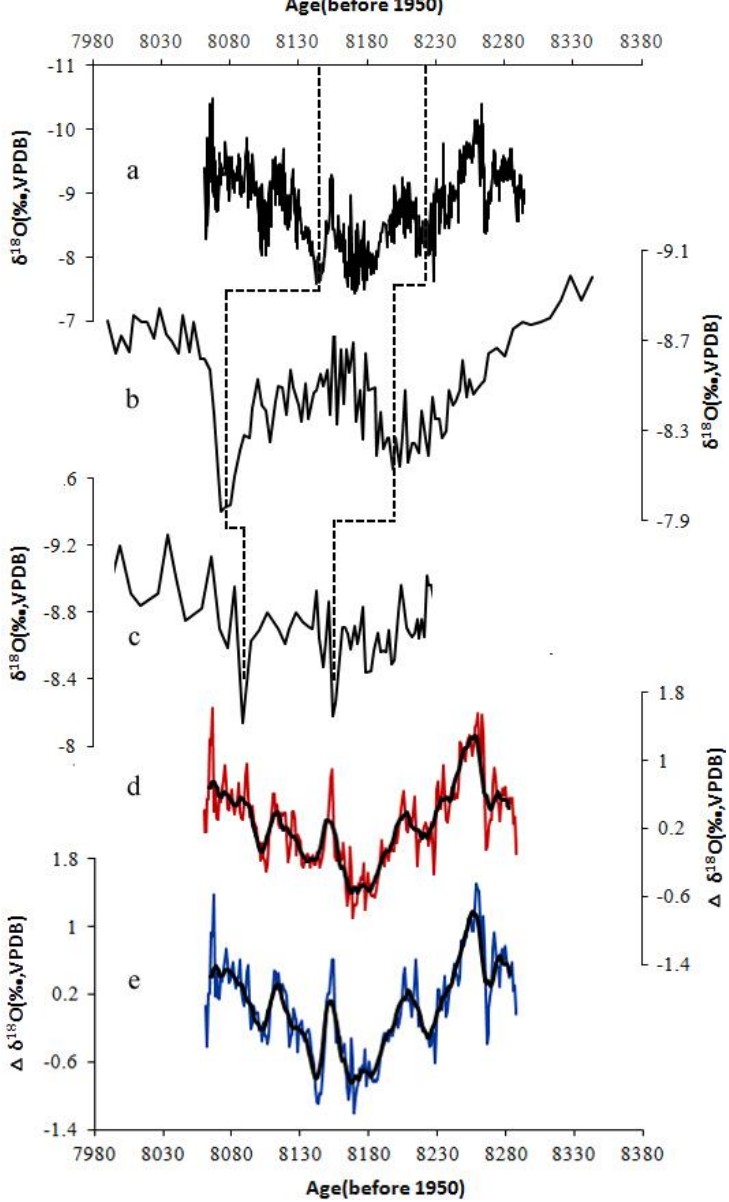


Figure 2. Original $\delta^{18}O$ stalagmite records adopted in this paper displayed with $\Delta\delta^{18}O$
sequences between stalagmites from Dongge and Heshang. a. HS4 $\delta^{18}O$ record from Heshang
cave(Liu et al., 2013); b. DA $\delta^{18}O$ record from Dongge cave(Cheng et al., 2009); c. D4 $\delta^{18}O$
record from Dongge cave (Cheng et al., 2009); d. $\Delta\delta^{18}O$ between DA and HS4 (red) with a 10-
year moving average(black); e. $\Delta$ $\delta^{18}O$ between D4 and HS4 (blue) with a 10-year moving
average (black). The dashed lines show matched peaks from each original record.

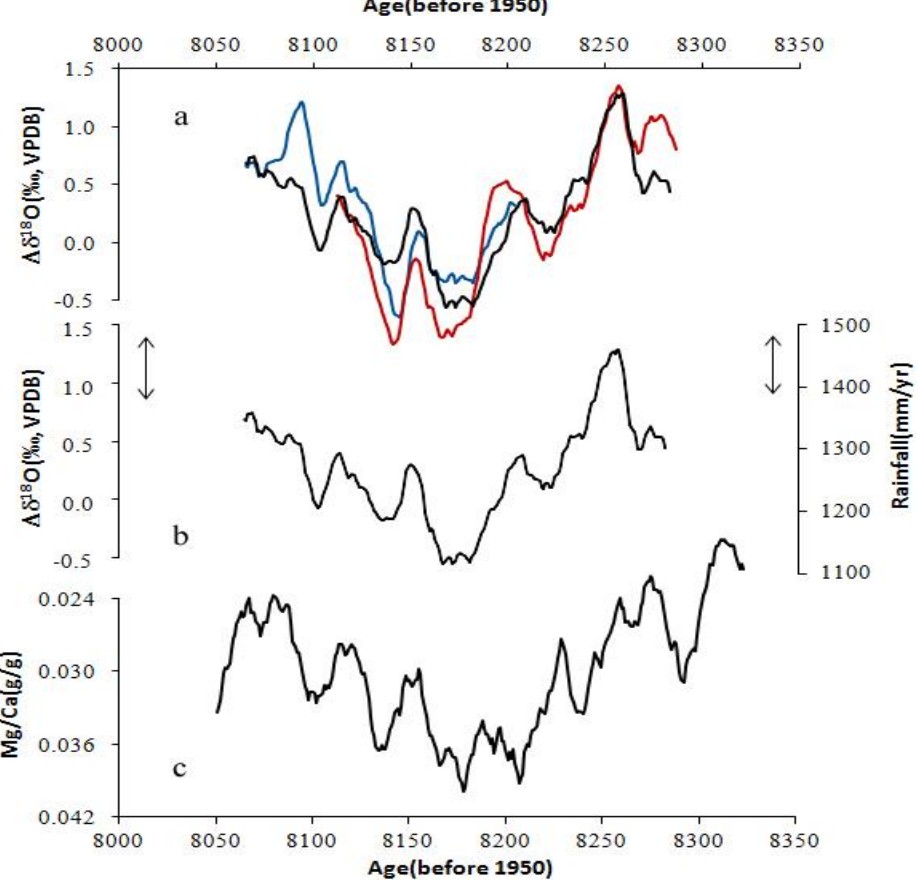


Figure 3. 10-yr moving average records during the 8.2 ka BP period. a. $\Delta\delta^{18}O$ records between
HS4 and DA with unchanged chronology (black), shifting DA 50-yr younger (blue) and 50-yr
older (red); b. $\Delta\delta^{18}O$ record between HS4 and DA and reconstructed annual rainfall in
southwest China with error bars indicated; c. Mg/Ca ratios of HS4 shown on inverted scales,
which reveals a similar trend to the rainfall sequence, increasing the confidence of the
quantization of the reconstructed record.

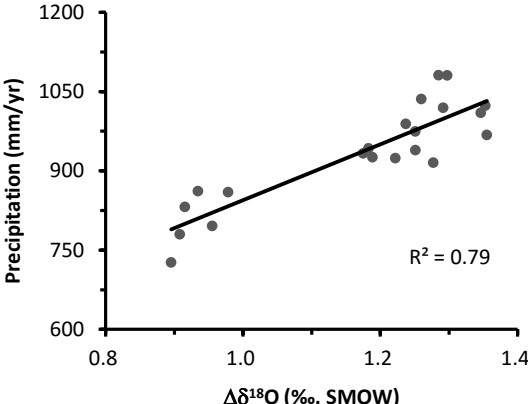


Figure 4. Correlation analysis between mean annual precipitation and drip water $\delta^{18}O$
difference from 2-month delayed LF6 and 4-month delayed HS4 from May 2011 to April
2014. The $\Delta\delta^{18}O$ data are calculated from monthly monitored data from Liangfeng cave and
Heshang cave (Duan et al., 2016). The annual precipitation data are the average from six sites
between Dongge cave and Heshang cave detailed in Hu et al. (2008a) and the original
monthly rainfall data are from http://www.wunderground.com/ history/. The correlation factor
of 0.79 indicates a significant positive correlation between regional annual rainfall and annual
$\Delta\delta^{18}$O.

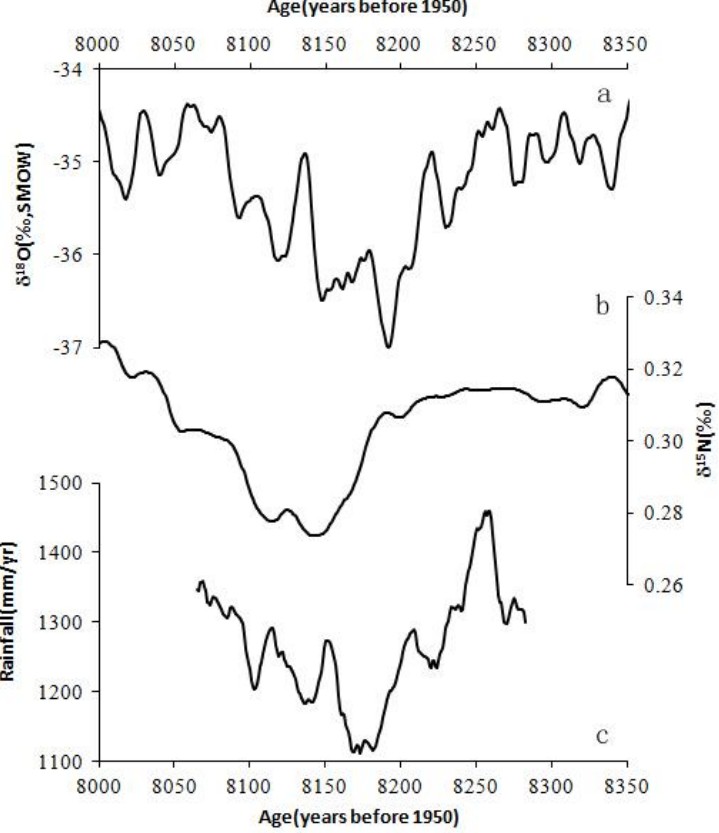


Figure 5. Records from Greenland ice core $\delta^{18}$O (Thomas et al., 2007) (a), Greenland ice core
$\delta^{15}$N (Kobashi et al., 2007) (b) and the reconstructed annual rainfall from this study(c) during
the 8.2 ka BP event. Three sequences show a similar pattern indicating the decrease in rainfall
in southwest China was coupled with Greenland cooling during the 8.2 ka BP event.