# Peer review of "Quantification of southwest China rainfall during the"

_Climate of the Past, 2015_

## Referee Comment (RC1) · Anonymous Referee #1 · 21 Feb 2016

This paper reconstructs rainfall variation in southwest China during the 8.2ka BP event by comparing Heshang cave $\delta$18O record with Dongge cave $\delta$18O record. The main method is similar to that in the paper "Hu et al., 2008 (EPSL)". Using this method, one important hypothesis is that Heshang cave and Dongge cave are in the same moisture transport pathway and the precipitation $\delta$18O difference between the two caves is mostly effected by the variation of precipitation amount. In the paper "Hu et al., 2008 (EPSL)", they considered that the two caves are in a uniform moisture transport pathway by using analysis of inter-annual variation in moisture transport during the instrumental record from 1952 to 2001. However, to our knowledge, the factors of stalagmite $\delta$18O at different timescales in monsoon area are very complex. The authors should

demonstrate that the stalagmite $\delta18O$ difference between the two caves is influenced by the variation of precipitation amount, by comparing the differences of precipitation amount, precipitation $\delta18O$, and stalagmite $\delta18O$ between the two caves. Because this is a critical assumption for this paper. As far as I know, some monitoring studies are going on in Heshang cave and Dongge cave during the past few years. I suggest the authors to verify the relation among the precipitation amount, precipitation $\delta18O$ and stalagmite $\delta18O$ by using modern monitoring data from the two caves. I think this manuscript should be published after revision.

———————————————————

---

## Referee Comment (RC2) · Anonymous Referee #2 · 27 Feb 2016

This paper presents a reconstruction of rainfall change during the 8.2 ka event based on taking the difference in measured d18O values from two caves supposedly along the same moisture transport pathway in China. Quantitative reconstructions are rare for this time interval, and it is a worthy goal to generate them. However, there are some major concerns about how this reconstruction is being created, including whether the d18O is a good indicator of precipitation amount and the selection of more robust methodologies of differencing two records with chronological uncertainties and different temporal resolutions. Lastly, it is unclear that the analysis computing the scaling of rainfall to Greenland temperature contributes much to our understanding.

Major comments:

1. Line 16: "decreased by ∼350 mm" This difference is calculated from a short-lived wet period occurring right before the 8.2 ka event. Rather, a longer-term average of pre-8.2ka conditions should be used to calculate this anomaly.

2. Since the publication of the Hu et al. (2008) paper, several other papers have been published showing that the relationship between d18O and precipitation amount is more complicated than assumed by the authors for their reconstruction. More consideration and discussion of these other results is needed, please see Liu et al. 2014 Quaternary Science Reviews 83: 115-128 and references cited therein.

3. Lines 87-88: Was this wiggle matching always within the analytical error of the U-Th dates?

4. Lines 114-117: A perhaps even larger source of error that could create negative values is the chronological uncertainty, given that two records with uncertain chronologies are being differenced. Wiggle matching will not eliminate this uncertainty, nor is even the best approach since it is subjective. Chronological error should be tracked in the reconstruction process.

5. How was a one-year resolution record created from a 2.5 year resolution record? Linear interpolation? A better approach would be to create records of equivalent >=2.5 year resolution.

6. The analysis of Yichang precipitation and Greenland temperature is not useful to the paper. It is unsurprising that the correlation of rainfall in China to temperature during the 8.2 ka event (perhaps the largest climate event of the Holocene) is larger than for interannual variations today calculated from two noisy station records. Regarding the calculated slopes of precipitation change per Greenland temperature change from the modern data, are these slopes shown to be significantly different than zero using a statistical test? This analysis is problematic in many regards, does not provide insight into "abrupt climate prediction under warming conditions" and should not appear in the paper.

Minor comments:

1. Line 32-33: The statement "experiencing a warming period similar to that of today" is debatable. There are important ways in which the early Holocene was different from today (e.g., melting of the Laurentide Ice Sheet, lower atmospheric carbon dioxide levels, etc).

2. Line 165-166: "highest annual rainfall of 350 mm/yr" This should read "maximum decline in annual rainfall of 350 mm/yr"

---

## Author Comment (AC1) · 29 Mar 2016

Referee #1

This paper reconstructs rainfall variation in southwest China during the 8.2ka BP event by comparing Heshang cave $\delta$18O record with Dongge cave $\delta$18O record. The main method is similar to that in the paper "Hu et al., 2008 (EPSL)". Using this method, one important hypothesis is that Heshang cave and Dongge cave are in the same moisture transport pathway and the precipitation $\delta$18O difference between the two caves is mostly affected by the variation of precipitation amount. In the paper "Hu et al., 2008(EPSL)", they considered that the two caves are in a uniform moisture transport pathway by using analysis of inter-annual variation in moisture transport during the

instrumental record from 1952 to 2001. However, to our knowledge, the factors of stalagmite δ18O at different timescales in monsoon area are very complex. The authors should demonstrate that the stalagmite δ18O difference between the two caves is influenced by the variation of precipitation amount, by comparing the differences of precipitation amount, precipitation δ18O, and stalagmite δ18O between the two caves. Because this is a critical assumption for this paper. As far as I know, some monitoring studies are going on in Heshang cave and Dongge cave during the past few years. I suggest the authors to verify the relation among the precipitation amount, precipitation δ18O and stalagmite δ18O by using modern monitoring data from the two caves. I think this manuscript should be published after revision.

Response

We do agree that the factors affecting stalagmite δ18O at different timescales in this monsoon area are very complex, and modern monitoring data from both Heshang Cave and Dongge Cave would be helpful to assess the δ18O difference method used in this study. Unfortunately, so far, there is no published cave monitoring data from Dongge Cave.

However, another cave located in Guizhou province with published monitoring records, named Liangfeng Cave (26°16'N, 108°03'E), might provide some useful information. There are three separate monthly drip-water δ18O data sets from April 2011 to April 2013 from Liangfeng (Zeng et al., 2015). To avoid evaporation influences, we chose the lowest δ18O value of each month to build a new δ18O sequence. Because of the aquifer above Heshang cave, the drip-water δ18O at HS4 collection site lags behind local rainfall δ18O by at least 1 month or even longer (Johnson et al., 2006). Therefore to establish a difference sequence between Liangfeng drip-water δ18O and HS4 drip-water δ18O, HS4 data is positively offset by two months to analyze the relation between the local rainfall amount and the drip-water Dδ18O.

Fig. S1 shows that there is a weak positive correlation (R=0.33) between monthly dripwater D$\delta$18O and average monthly rainfall amount from 6 sites mentioned in Hu et al.(2008a). As stalagmite $\delta$18O derives from cave drip water $\delta$18O, in some degree the weak positive correlation shown in Fig. S1 suggests that stalagmite $\delta$18O differences between two caves located along the same moisture transport pathway could reflect the local rainfall amount.

Relevant revision will be done in the manuscript.

———————————————————

[Figure]

Figure S1. Correlation analysis between monthly drip-water difference of Liangfeng Cave(Zeng et al.,2015) and HS4 collection site and average local monthly rainfall amount from April 2011 to April 2013. Monthly average rainfall data are from instrumental records (http://www. wunderground.com/ history/wmo/) of 6 sites mentioned in Hu et al.(2008), while $\Delta\delta^{18}O$ is from the difference between Liangfeng monthly cave drip-water $\delta^{18}O$ and HS4 drip-water $\delta^{18}O$ being positively offset by two months.

**Fig. 1.**

---

## Author Comment (AC2) · 29 Mar 2016

Reviewer #2

Major comments:

1. Line 16: "decreased by âĹij350 mm" This difference is calculated from a short-lived wet period occurring right before the 8.2 ka event. Rather, a longer-term average of pre-8.2ka conditions should be used to calculate this anomaly.

Response

If calculated from a longer-term average, between the average value before the event

and the average value during the event, the difference would be ~140 mm, which is similar to the difference shown in Hu et al.(2008). Since the record resolution of Hu et al.(2008) is ~100 yr, which is difficult to show the abrupt decrease when the 8.2 kyr event occurred, we therefore prefer to take the advantage of the high resolution of this record to show the abrupt change.

2. Since the publication of the Hu et al. (2008) paper, several other papers have been published showing that the relationship between $\delta$18O and precipitation amount is more complicated than assumed by the authors for their reconstruction. More consideration and discussion of these other results is needed, please see Liu et al. 2014 Quaternary Science Reviews 83: 115-128 and references cited therein.

Response

We do agree that the interpretation of Chinese stalagmite $\delta$18O is complex, and more discussion shown as follows will be added in the manuscript.

Many processes contribute to Chinese stalagmite $\delta$18O, such as moisture source and pathway, local condensation and evaporation or even different types of precipitation (Dayem et al., 2010). A recent millennial climate simulation suggests that the Chinese stalagmite $\delta$18O record is an indicator of intensity of the East Asian summer monsoon in terms of the monsoon wind and the accompanying rainfall in northern China, but not related to the rainfall change in southeastern China (Liu et al., 2014).

Since stalagmite $\delta$18O records from South China are more complex, modern monitoring data from both Dongge and Heshang might be helpful to assess the difference method adopted in this paper. Unfortunately there is no published monitoring data from Dongge, but there are three separate monthly drip-water $\delta$18O records from Liangfeng Cave (26°16'N, 108°03'E, close to Dongge Cave) from April 2011 to April 2013(Zeng et al., 2015). To avoid the effect of evaporation, we selected the lowest $\delta$18O value from each month from Liangfeng to calculate the drip-water $\delta$18O difference between Liangfeng and HS4 collection site after the whole HS4 monthly drip-water $\delta$18O record

was positively offset by 2 months to allow for the effect of the aquifer above Heshang cave(Johnson et al., 2006). Correlation analysis suggest that there may be a weak positive correlation(R=0.33) between the monthly drip-water D$\delta$18O and the average monthly rainfall from 6 sites mentioned in Hu et al. (2008a). Since stalagmite $\delta$18O derives from cave drip-water $\delta$18O, in some degree this weak correlation between the cave drip-water D$\delta$18O and the local rainfall amount suggests that stalagmite D$\delta$18O between two caves located along the same moisture transport pathway could reflect the local rainfall.

3. Lines 87-88: Was this wiggle matching always within the analytical error of the U-Th dates?

Response

Yes. From the chronology table of stalagmite DA (Cheng et al., 2009), the errors of the chronology of DA during 8.2 kyr period is from 31-yr to 94-yr with an average of ∼60-yr. Since the difference between adjusted and original chronology of DA is from 2-yr to 70-yr with an average of ∼40-yr, we are sure that the wiggle of DA is always within the analytical uncertainty of its U-Th dates.

4. Lines 114-117: A perhaps even larger source of error that could create negative values is the chronological uncertainty, given that two records with uncertain chronologies are being differenced. Wiggle matching will not eliminate this uncertainty, nor is even the best approach since it is subjective. Chronological error should be tracked in the reconstruction process.

Response

Yes, we do agree that the chronological error should be tracked in the reconstruction process. We tested this by shifting the DA $\delta$18O data set by moving 50-yr forward and backward respectively as shown in revised Figure 2. After the shifting, though it does increase the uncertainty of the D$\delta$18O with a maximum error of 0.76‰ the

general variation trends are similar, suggesting the difference method is valid in this case. However, the error produced by chronology uncertainty should be taken into consideration. Therefore the cumulative error of the reconstructed D$\delta$18O sequence should increases to 0.53‰

The relevant correction will be done in the manuscript.

5. How was a one-year resolution record created from a 2.5 year resolution record? Linear interpolation? A better approach would be to create records of equivalent >=2.5 year resolution.

Response

We rechecked the $\delta$18O records of DA from Cheng et al.(2009), and the resolution varies from 1-yr to 8-yr with an average of ~3.5yr. That means even if we create an HS4 $\delta$18O records with a 2.5-yr or 3.5-yr resolution, it is still difficult to be equivalent to DA. Therefore, annual interpolation is perhaps the best way to make the two $\delta$18O sequences comparable.

6. The analysis of Yichang precipitation and Greenland temperature is not useful to the paper. It is unsurprising that the correlation of rainfall in China to temperature during the 8.2 ka event (perhaps the largest climate event of the Holocene) is larger than for interannual variations today calculated from two noisy station records. Regarding the calculated slopes of precipitation change per Greenland temperature change from the modern data, are these slopes shown to be significantly different than zero using a statistical test? This analysis is problematic in many regards, does not provide insight into "abrupt climate prediction under warming conditions" and should not appear in the paper.

Response

We will delete the discussion section about the analysis of Yichang precipitation and Greenland temperature.

[Figure]

Minor comments: 1. Line 32-33: The statement "experiencing a warming period similar to that of today" is debatable. There are important ways in which the early Holocene was different from today (e.g., melting of the Laurentide Ice Sheet, lower atmospheric carbon dioxide levels, etc).

Response

We will delete this sentence.

2. Line 165-166: "highest annual rainfall of 350 mm/yr" This should read "maximum decline in annual rainfall of 350 mm/yr"

Response

This will be corrected.

[Figure]

[Figure]

Figure 2. 10-yr moving average records during the 8.2 ka BP period. a) Δδ¹⁸O records between HS4 and DA with unchanged chronology(black), 50-yr moving forward (blue) and 50-yr moving backward(red); b) Δδ¹⁸O records between HS4 and DA and reconstructed annual rainfall in southwest China with their accumulative error bars ; c) Mg/Ca ratios of HS4 shown on inverted scales, which reveals a similar trend to the rainfall sequence increasing the confidence of the quantization of the reconstructed record.

**Fig. 1.**

---

## Author Response (AR1)

**Responses to the Editor**

*Initial comments in red italics*, **responses in plain text**

*Both referees have major concerns regarding the interpretation of the calcite oxygen isotopes and both are asking for more detailed information on oxygen isotopes in modern precipitation. I support their concerns and a more detailed discussion on the climatic and environmental factors influencing del18O in present-day rainfall is necessary to warrant publication in CoP. Furthermore, I wonder whether you could also provide further information on uncertainties of your quantitative reconstruction.*

1)**We put more information about the calcite oxygen isotopes in the revised manuscript shown as follows (line 127-138). The idea of reconstructing regional rainfall between two caves by comparing two spatially separated cave records along the same moisture transport pathway is to presume single stalagmite $\delta^{18}$O values from monsoon areas at least contain rainfall information. For Chinese stalagmite $\delta^{18}$O values, they are indeed influenced by different types of precipitation, and as well as moisture source and its pathway, local condensation and evaporation processes (Dayem et al., 2010). And a recent millennial climate simulation also suggests that Chinese stalagmite $\delta^{18}$O records could be used as an indicator of intensity of the East Asian summer monsoon in terms of the continental scale Asian monsoon rainfall response in the upstream regions (Liu et al., 2014). As both Dongge and Heshang $\delta^{18}$O records respond to the upstream rainfall respectively, the difference of the two records should be related to the regional rainfall between Dongge and Heshang cave.**

2)**As compared with oxygen isotopes in modern precipitation, monitoring cave drip water $\delta^{18}$O should reflect stalagmite $\delta^{18}$O more directly. Though there are no published monitoring records from Dongge cave, we pick up a drip water $\delta^{18}$O record from May 2011 to April 2014 from Liangfeng cave, a cave close to Dongge, to compare the drip water $\delta^{18}$O data between Liangfeng and Heshang. A significant positive correlation (R$^2$=0.79) between annual drip water $\Delta\delta^{18}$O$_{LF-HS}$ and regional annual rainfall amount gives a modern support for the reconstruction method. More details are shown from line 139 to line 180 in the revised manuscript.**

3)**To better access the uncertainty of the $\Delta\delta^{18}$O record, chronology uncertainty has been discussed in the revised manuscript, which produces a maximum error of 0.76‰. Taken all the factors into consideration, the final uncertainty of $\Delta\delta^{18}$O would be ~0.53‰, therefore the uncertainty of the reconstructed rainfall in southwest China would be ~100 mm/yr. The details are shown from line 101 to line 111, line 119 and line 126.**

4)**According to all the comments from the anonymous reviewers and the editor, we make a major revision on the manuscript by deleting the discussion section about the analysis of Yichang precipitation and Greenland temperature and the original Fig. 4, restructuring Method section with more discussions on the uncertainty of the $\Delta\delta^{18}$O record and with more modern monitoring supports for the reconstruction method, revising Fig. 2 and adding another figure shown as Fig. 3. All relevant revised parts are marked in red.**

---

## Author Response (AR2)

**Responses to the Editor**

*Initial comments in red italics*, **responses in plain text**

*I would like to thank you for your careful revisions. However, I think there are some additional revisions necessary. First of all, I would strongly recommend to find a native speaker to proofread your manuscript. There are many examples of linguistic defficiencies and grammatical inconsistencies. Furthermore, I would suggest strongly to include a map showing all the cave sites used for this study.*

1) **Dr. Nick Belshaw, a native speaker, from Earth Science Department, Oxford University, helped with the manuscript proofreading. And all corrections are shown in the marked-up manuscript version. The deleted words are shown as blue-colored texts with a line in the middle, while the inserted words are shown as red-colored texts.**

2) **A map with all the cave sites used for this study has been included in the revised manuscript shown as Fig. 1.**

[revised manuscript text omitted]

(2008a) and the original monthly rainfall data are from http://www.wunderground.com/
history/. The correlation factor of 0.79 indicates a significant positive correlation between
regional annual rainfall and annual $\Delta\delta^{18}O$.

[Figure]

Figure 5. Records from Greenland ice core $\delta^{18}O$ (Thomas et al., 2007) (a), Greenland ice core
$\delta^{15}N$ (Kobashi et al., 2007) (b) and the reconstructed annual rainfall from this study(c) during
the 8.2 ka BP event. Three sequences show a similar pattern indicating the decrease in rainfall
in southwest China was coupled with Greenland cooling during the 8.2 ka BP event.